# How Loneliness Worked on Suicidal Ideation among Chinese Nursing Home Residents: Roles of Depressive Symptoms and Resilience

**DOI:** 10.3390/ijerph18105472

**Published:** 2021-05-20

**Authors:** Yang Yang, Rui Wang, Dan Zhang, Xia Zhao, Yonggang Su

**Affiliations:** 1School of Nursing and Rehabilitation, Shandong University, Jinan 250012, China; hlyangyang@sdu.edu.cn (Y.Y.); wangr164042135@163.com (R.W.); zhangdan2014sdu@163.com (D.Z.); 2Department of Health Management, Heze Medical College, Heze 274000, China; zhaoxia9412@163.com; 3School of Basic Medical Sciences, Shandong University, Jinan 250012, China; 4School of Foreign Languages and Literature, Shandong University, Jinan 250012, China

**Keywords:** suicidal ideation, resilience, loneliness, depressive symptoms, nursing home residents

## Abstract

Suicide in later life is becoming severe under rapid population aging, especially for nursing home residents. Loneliness, an increasingly represented issue among nursing home residents, is found to be a risk factor for depressive symptoms. Both loneliness and depressive symptoms may contribute to the development of suicidal ideation. According to the Protective Factor Model, resilience can act as a moderating role interacting with risk factors to buffer the negative effects on the outcome. The present study aimed to assess the mediating role of depressive symptoms and the moderating effect of resilience on the risk factors of suicidal ideation to attenuate the adverse contribution among nursing home residents. A total of 538 nursing home residents participated in this cross-sectional study, and their suicidal ideation, resilience, loneliness, and depressive symptoms were measured. The mediating effect and moderated mediation model were tested using the Macro Process of SPSS 21.0. Statistics showed that 19.7% of participants had suicidal ideation. The mediating model (H1: *B* = 0.477, *p* < 0.001; H2: *B* = 0.325, *p* < 0.001; H3: *B* = 0.308, *p* < 0.001) and the moderating effect of resilience interacting loneliness (H4: *B* = −0.133, *p* < 0.001; H6: *B* = −0.109, *p* < 0.001) and depressive symptoms (H5: *B* = −0.077, *p* < 0.001) were statistically significant. The findings indicated the protective effect of resilience in alleviating the negative influence of risk factors for suicidal ideation, suggesting that positive psychological interventions for resilience building might be effective in suicide prevention among nursing home residents.

## 1. Introduction

Later-life suicide has been an intractable problem for clinicians worldwide [1] The statistics of suicide in older adults, as reported by the World Health Organization [2], were much higher than those in the younger population. Moreover, for every older adult who died by suicide, it was estimated there were four who attempted suicide [3]. Over recent decades, China has witnessed rapid population aging and stepped into the advanced stage of aging characterized by disability and multimorbidity. Among 253.88 million Chinese older adults aged 60 and above, over 2 million currently reside in nursing homes [4]. Under this circumstance, nursing home residents are comparatively older, with a lower level of physical function and severer psychological disorders when compared to their community-dwelling counterparts [5]. Influenced by traditional Chinese culture and specifically Confucian values of filial piety and family unity, older adults tend to attach negative stereotypes to living in nursing homes, interpreting it as an unfilial sign of children and limited mobility and socialization on the part of themselves or developing a sense of being deserted [6]. Moreover, as the social life of Chinese older adults is mainly family-oriented with kin relationships as the center, nursing homes fail to provide elderly residents with a sense of belonging [7]. As reported from sparse studies, nursing home residents with limited social activities and diagnosis of depressive symptoms suffer a higher risk of suicide [8,9]. On the basis of this, it is pressing to address these factors as a prior concern, especially for nursing home residents.

Suicidal ideation can be defined as suicide-related thoughts, intentions, and plans [10]. It has been revealed as a major predictor for suicidal behavior in the sense that approximately one-third of individuals with suicidal ideation finally developed suicidal implementers [11]. According to the Interpersonal-Psychological Theory of Suicide (IPTS), perceived burdensomeness and thwarted belonging are two active psychological constructions that can predict suicidal ideation [12]. It may be feasible to apply IPTS to explore the influencing factors of suicidal ideation among nursing home residents. Although suicidal ideation and its related factors have been widely explored among adolescents both from perspectives of physiology and psychology [13,14], little evidence exists among the sample of nursing home residents, a group of high risk for suicidal ideation. Thus, understanding the relevant factors contributing to suicidal ideation is critical towards the preliminary prevention of suicide among older residents in nursing homes.

Loneliness has been primarily identified as a major risk factor for older suicides, reflecting a subjective state of lacking desired affection or of insufficient closeness to friends and families. As an important indicator for measuring social well-being and the quality of life of older individuals, loneliness has become a prominent problem threatening psychological health and is also associated with premature mortality [15,16]. Studies reveal an increasing prevalence of loneliness at advanced ages, especially with those in institutional care homes [17]. The dimension of “thwarted belongingness” from IPTS suggests that loneliness can lead to self-destructive behavior [18], whereas the assumption that the development of influencing factors to suicidal ideation in all groups of people will be the same is problematic. Similarly, as a possible risk factor for depressive symptoms, loneliness has been shown to prospectively predict suicide attempts through mediation as well [19]. Therefore, it might be plausible that loneliness raises the incidence of suicidal ideation by increasing the risk of depressive symptoms. Depressive symptoms, a robust predictor of suicide, turned out to be highly prevalent in a former study, at 16.7–40.3% [5], of Chinese nursing home residents. A population-based cohort [20] regards depressive symptoms as the predictor of suicidal ideation, as it includes perceived burdensomeness, a positive psychological construct of IPTS. However, the interaction of thwarted belongingness and perceived burdensomeness does not significantly predict suicidal ideation [21], suggesting that the two predictors for suicidal ideation may have other relationships. Research conducted in patients with chronic pain and older adults (with declining vitality) has proved the mediating effect of depressive symptoms [22,23], suggesting its potential effect in mediating risk factors and suicidal ideation among the nursing home residents. According to the Protective Factor Model of Resilience [24], the negative effect of risk factors on the outcome can be alleviated by protective factors through interaction. Hence, given the potentially higher risk of suicidal ideation among nursing home residents, it is urgent and significant to explore the protective factors that may buffer the negative impact of risk factors.

Resilience is a psychological construct, acting as an ability or perseverance of an individual to overcome difficulties or as a personal resource to buffer the individual against adversity [25]. As a micro-level protective factor of suicidal ideation, resilience is of great utility in coping with adverse emotional situations and initiating positive adaptation [26]. Previous studies indicated that resilience was obviously counteractive with loneliness and depressive symptoms [27,28], suggesting that resilience might attenuate the strength of the association between high-risk factors and suicidal ideation. As stated by the Protective Factor Model of Resilience [29], there is an interaction between protection and risk factors, which reduces the probability of negative outcomes and moderates the effect of exposure [30]. On this basis, resilience may act as a moderator, rendering the correlation between risk and suicide to a lower or nonexistent level. Nowadays, academic interest in the moderating effect of resilience on suicide, especially with psychiatric patients and adolescents [31], is increasing [25,32], whereas that in nursing home residents is rare. Although former research supports the importance of resilience against suicide [33,34], research on its influence on samples from older adults is still inadequate, and the interaction mechanism of resilience as a moderator on suicide risk factors among nursing home residents is not clear.

Thus, the current study aims to explore the moderating effect of resilience on suicidal ideation through buffering the impact of loneliness and depressive symptoms. Our hypothesis is that loneliness may develop into suicidal ideation either directly (H3) or through the mediating role of depressive symptoms (H1, H2), and the hypothetical paths can be moderated by resilience (H4–H6). Six paths have been tested to verify the association among variables (See Figure 1).

## 2. Materials and Methods

### 2.1. Study Design

This cross-sectional, observational study was conducted under the guidance of the STROBE statement, involving 37 nursing homes from September 2018 to April 2019 in Jinan city of eastern China.

### 2.2. Setting and Participants

Data were collected using stratified random sampling. As the second populous province in China, Shandong is a typical northern province with the largest aging population. By the end of 2019, approximately 23.2521 million older adults aged over 60 resided in Shandong, accounting for 23.09% of the aged people countrywide [35]. Jinan, the provincial capital city, ranked fourth in Shandong, with a registered population of nearly 7.96 million [36]. Around 16.22% (6/37) of nursing homes were public facilities affiliated with public hospitals or social welfare institutions, and the rest were profit-making entities, in a proportion of more than 20% of the total [37]. By estimation, a total of ten thousand beds were offered in all these nursing homes with an occupancy rate of about 60%.

Nursing homes from 7 districts (i.e., Lixia, Huaiyin, Tianqiao, Shizhong, Licheng, Changqing, and Zhangqiu district), with a capacity of over one hundred beds and in an occupancy rate of over 60%, were first selected according to the different levels of economic development of each district in the target city, resulting in 37 nursing homes recruited accounting for 30% of the total number. Residents in chosen nursing homes were approached if they fulfilled the inclusion criteria: (1) aged 60 and above [38], (2) living in an institution for at least one month [39], and (3) capable of verbal communication. Residents with severe cognitive impairment (the scores of Mini-Mental State Examination (MMSE) ≤ 9 [40]), in psychotherapy or in active illness with acute symptoms (pain, nausea, etc.) or with terminal illnesses, were excluded [41]. Among 562 nursing home residents meeting the criteria, 21 refused, and 3 were excluded (i.e., one each for dementia, cognitive impairment, and hearing loss), and a total of 538 older adults eventually completed this study without attrition.

This study was approved by the Shandong University Human Research Protections Program. All participants willing to participate in the study signed informed consent before the investigation. Among those who were illiterate or unable to write, their families or institution manager signed on their behalf after receiving oral consent.

### 2.3. Covariates

Sociodemographic covariates included gender (0 = female, 1 = male), age (60–94), marital status (1 = unmarried, 2 = married, 3 = divorced, 4 = widowed), education (1 = illiterate, 2 = primary school, 3 = junior high school, 4 = senior high/above), self-rated financial status (1 = good, 2 = medium, 3 = poor), and children visit frequency (1 = once per 1–2 weeks, 2 = once per over 2 weeks). 

According to our previous study conducted in Chinese rural nursing homes [42], physical problems can indirectly affect suicidal ideation. Therefore, it was determined that such health-related covariates be included as cognitive function and comorbidities, which were measured by the MMSE and Medical Disorders (MD) scale.

### 2.4. Measures

#### 2.4.1. Comorbidities

Twelve common physical disorders [43] were listed for the participants to choose from, including diabetes, hypertension, osteoarthritis, liver disorders, kidney disorders, cancer, congestive heart failure, chronic obstructive pulmonary disease, heart attack, gastrointestinal disorders, hearing loss, and ophthalmologic diseases. Each disorder is scored as one point, and the scale score is the sum of the number of specified disorders.

#### 2.4.2. Cognitive Function

The Mini-Mental State Examination, developed by Fostein et al. [40], is a practical method of grading cognitive impairment, with 5 domains covering orientation, registration, attention and calculation, memory, and linguistic competence [44]. The Chinese version of MMSE we used was externally validated for local use in a population-based study conducted in Chinese older adults [45]. This instrument was coded according to previous conventions: items were coded as zero if the respondents refused or were unable to complete. The total score ranged from 0 to 30, with higher scores signifying better cognitive function, and scores of ≤24 indicating impaired cognitive function of older adults [46]. The Cronbach’s alpha in this study is 0.759.

#### 2.4.3. Loneliness

The third version of the UCLA loneliness scale (ULS) was used to measure the levels of loneliness [47], which has been validated in studies conducted in several languages, including Chinese [48]. A 0.942 Cronbach’s alpha was obtained in the sample of Chinese rural nursing home residents [42]. The scale contained 20 items, with each item scored on a scale from 1 (“never”) to 4 (“always”). The total score of the scale ranged from 20 to 80, a high level of loneliness being confirmed with a score of ≥44 and higher scores denoting a severer feeling of loneliness. The Cronbach’s alpha for the current sample was 0.964.

#### 2.4.4. Depressive Symptoms

Depressive symptoms were measured using the Hospital Depression Scale (HDS) and a 7-item depression subscale of Hospital Anxiety and Depression (HAD). Despite the wide application in psychiatric patients, HDS has been found to be applicable for the assessment of depressive symptoms in nonclinical settings such as nursing homes [49,50]. The original English version has been translated into Chinese and presented good reliability and validity among Chinese nursing home residents [42]. The total score was 21 on a 4-point Likert scale, with higher scores indicating severer depressive symptoms. An HDS score < 8 meant no depressive symptoms, while an HDS score of ≥11 suggested a major depressive disorder requiring further clinical investigation. The Cronbach’s alpha for the current sample was 0.877.

#### 2.4.5. Resilience

The 10-item Connor–Davidson Resilience Scale (CD-RISC-10), originating from the 25-item CD-RISC developed by Connor and Davidson [51], proved to be a well-validated measurement in the general population. Based on the modification and examination by Campbell and Stein [52], the shortened version was obtained, which displayed excellent psychometric properties among undergraduates. The Chinese version was translated by Ye et al [53] and examined in the sample of parents group of children with cancer. Wu et al [54] adopted this scale of the Chinese version in the sample of rural nursing homes and validated a Cronbach’s alpha of 0.97 with an intraclass correlation of 0.97. The CD-RISC-10 contained “hardiness” factor (8 items), i.e., the ability to cope with change, unexpected accidents, stress, illness/hardship, pressure, negative outcomes, and negative emotions, and “persistence” factor (2 items), i.e., belief in one’s ability to achieve goals despite obstacles and not giving up. Response options were scored on a 5-point frequency Likert scale (0 = never; 1 = rarely; 2 = sometimes; 3 = usually; 4 = always), generating a range of scores from 0 to 40, with higher scores signifying a higher level of resilience. The Cronbach’s alpha in this study was 0.970.

#### 2.4.6. Suicidal Ideation (SI)

The Chinese version of the 19-item Beck Suicide Ideation Scale (BSI-CV) was translated by Li et al. [55] and has shown good reliability and validity among Chinese adult community residents, in both urban and rural areas. The scale contains a suicidal ideation screen for 5 items and suicidal tendency for14 items [56]. Respondents who answered item 4 (i.e., active suicidal ideation) or item 5 (i.e., passive suicidal ideation) with “weak” or “moderate to strong” (i.e., not 0) were considered as “suicidal ideator” and, in turn, requested to continue the remaining items 6–19. The total scores are 0 to 38, with higher scores indicating stronger suicidal ideation. The Cronbach’s alpha of this study is 0.919.

### 2.5. Data Collection and Control

Participants recruited in this study were required to answer every question of each scale during the survey conducted in their separate rooms. Data were collected by trained postgraduates in the researchers’ university. To minimize the likelihood of missing data, investigators checked and confirmed all the options with older adults on the spot for each completed questionnaire so that there were no missing data in this study. To minimize reporting bias, the investigators were trained to interview participants in a consistent way and were not allowed to guide them to respond.

### 2.6. Statistical Analysis

All analysis was carried out with SPSS 21.0 software (IBM, Armonk, NY, USA). Statistical significance was defined as a two-tailed value of *p* < 0.05. Basic characteristics of participants were illustrated by descriptive analyses, and Pearson Correlation Analysis was employed to examine the association among suicidal ideation, resilience, loneliness, and depressive symptoms.

The two-step statistical analyses were conducted through the SPSS PROCESS Macro version 3.0 [57] to test the mediating effect and moderated mediation effect. The mediating role of depressive symptoms was examined using Model 4, and then Model 59 was employed for the moderated mediation effect. The 95% confidence interval (CI) was calculated with 5000 bootstrap resamples. If 0 was not in the 95% CI, a significant model could be established. Basic characteristics of participants (gender, marital status, education, self-rated financial status, number of physical diseases, and MMSE) were controlled as covariates in all models and the study variables were standardized. 

The Johnson–Neyman technique refers to the method that aims to seek the values of the moderator with the simple slope of the dependent variable regressed on the independent variable as significant [58]. According to Hayes and Rockwood [59], the J–N technique analytically derives the values of the moderator that identifies points of transition along the continuum of the moderator between a statistically significant and nonsignificant effect of the independent variable. Continuously plotted confidence intervals around the simple slopes for all values of resilience are termed confidence bands [60]. When the moderator reaches a certain value where the confidence bands do not contain zero, the relationship between the independent variable and the dependent variable could not be established [58]. Therefore, to clearly show the conditional effect of loneliness and depressive symptoms at values of the moderating effect of resilience in the observed range, the Johnson–Neyman technique using the visualization method of PROCESS Macro was plotted.

## 3. Results

### 3.1. Basic Characteristics of Participants with or without Suicidal Ideation

Of 538 recruited older adults, 321 (59.67%) were female and 217 (40.33%) were male, aged 78.13 ± 8.72 (60–94). Among the entire sample, 32.7% (176) reported severe loneliness, and 14.9% (80) had suicidal ideation during the past week. More information is presented in Table 1. 

### 3.2. Bivariate Correlations between Main Variables

Means, SD, and correlations of variables are shown in Table 2. Loneliness was positively correlated with depressive symptoms (*r* = 0.532, *p* < 0.001) and suicidal ideation (*r* = 0.484, *p* < 0.001), whereas resilience was negatively linked to loneliness (*r* = −0.615, *p* < 0.001), depressive symptoms (*r* = −0.551, *p* < 0.001), and suicidal ideation (*r* = −0.471, *p* < 0.001).

### 3.3. Mediation Effect of Depressive Symptoms

The direct and indirect effects are presented in Table 3. In particular, the significant effect value of loneliness on depressive symptoms (H1) and suicidal ideation (H2) was 0.308 (95% CI: 0.200–0.416) and 0.477 (95% CI: 0.386–0.564), respectively, indicating the total effect being 0.463 (direct effect: 0.308, indirect effect: 0.155), among which the mediating effect of depressive symptoms accounted for 33.48%.

### 3.4. Moderated Mediation Effect of Resilience

As presented in Table 4, the interaction of loneliness and depressive symptoms with resilience was significant (H4: Loneliness × Resilience: B = −0.133, 95% CI: −0.187–−0.079; H5: Depressive symptoms × Resilience: B = −0.077, 95% CI: −0.147–−0.007), indicating the indirect effect of loneliness and depressive symptoms on suicidal ideation was moderated by resilience. Moreover, Path H6 showed a significant moderating effect of resilience on the direct effect of loneliness on suicidal ideation (Loneliness × Resilience: B = −0.109, 95% CI: −0.177–−0.041), indicating the multiple moderating roles of resilience on the development paths of suicidal ideation.

The conditional effect of loneliness on suicidal ideation through depressive symptoms was moderated by different levels of resilience when it was low (*B* = 0.389, 95% CI: 0.253–0.525) and high (*B* = 0.049, 95% CI: −0.153–0.251). Specifically, a higher level of resilience predicted a weaker effect of loneliness and depressive symptoms on suicidal ideation. On this basis, the Johnson–Neyman technique was applied to indicate the specific value of resilience when it counteracted the direct and indirect effects of loneliness completely. As shown in Figure 2 and Figure 3, the effects of loneliness on depressive symptoms and suicidal ideation were completely moderated when resilience reached the value of 0.917 and 0.697, respectively. Similarly in Figure 4, when the standardized scores of resilience were higher than the point of 0.862, the confidence bands of depressive symptoms on suicidal ideation did not include 0. Even though the intercept points of resilience were presented towards the later of the simple slope, the nonsignificant confidence bands still suggested a considerable potential of resilience in moderating the association between loneliness, depressive symptoms, and suicidal ideation.

## 4. Discussion

The current study explored the development paths of loneliness and depressive symptoms to suicidal ideation and assessed the protective mechanism of resilience. Results supported the role of depressive symptoms in mediating loneliness to suicidal ideation and verified the buffering effect of resilience on the negative impact of risk factors on suicidal ideation.

Findings of this study showed the partial mediating role of depressive symptoms in the association between loneliness and suicidal ideation, suggesting a possible interpretation for the association between thwarted belongingness and perceived burdensomeness when predicting suicidal ideation among nursing home residents. A review of the exiguous literature on the attitudes towards nursing homes among Chinese older adults revealed that community-dwelling seniors regarded residing in nursing homes as “living in jail” [61]. Corresponding to such an attitude, some nursing home residents in a qualitative study expressed their suffering of loneliness and wish, or even pray, to return home [6], presenting a severely thwarted sense of belonging with nursing homes. Due to the increasing need for professional care caused by the decline in physical function, older adults relocated to nursing homes were likely to consider themselves a burden to their family members and were afraid of being abandoned [6,62]. Unsatisfied internal needs and an increasing sense of burden of seniors harm their self-esteem and trigger their hopelessness in life, attributing to a headstream of depressive symptoms [63,64]. According to the work plan of exploring feature services for the prevention and treatment of depression released by the National Health Commission [65], depression screening will be included in physical examinations for students in high schools, colleges, and universities. Comparably, it is highly recommended that in the future similar screening be carried out in nursing home residents for the primary prevention of suicide.

In addition, this study confirmed the moderating effect of resilience, revealing its protective mechanism against the risk factors of suicidal ideation among nursing home residents. The findings confirmed the Protective Factor Model that resilience could buffer the negative effect through interacting with the stressors of suicidality. For Chinese nursing home residents who are deeply influenced by family culture and filial piety, the concept that adults’ children’s obligation and responsibility are to take care of old parents has been deeply rooted in mind [66]. Relocation in nursing homes means a loss of former familiar surroundings, kinship, and social networks for Chinese older adults [62]. It may be practicable to further explore the positive internal resources to prevent suicide in Chinese older adults rather than to intervene the risk factors after problems show up. Resilience presents one’s psychological ability and the internal resource to counter decrease the risk of a negative outcome and increase the likelihood of positive outcome for individuals [67]. Resilience can be regarded as a dynamic process, featuring self-regulation and development across one’s lifetime [68]. Through the results, we can find that stronger resilience predicts a weaker negative impact of loneliness and depressive symptoms on suicidal ideation, suggesting the significance of resilience building in older adults. Previous studies on resilience intervention have mostly been conducted among children and adolescents in the framework of family units, which has shown to be effective in reducing stress and promoting connection [69]. For example, Henderson and Milstein [70] put forward the resilience training plan of six strategies, including: providing opportunities for students to participate in meaningful activities; establishing and maintaining high expectations for students; creating a caring and supportive school atmosphere; strengthen the prosocial tendency; establishing clear and consistent codes of conduct; teaching social and life skills. However, relevant evidence is lacking on resilience intervention in nursing home residents. The multiple moderating effects of resilience in our findings provide feasibility for the implementation of resilience intervention implemented in nursing home residents. Considering the lack of professional psychological consultants in a majority of Chinese nursing homes, caregivers, such as nursing staff, the most accessible to nursing home residents, can be trained with the resilience intervention techniques so as to supplement the resilience training for older adults in their daily care. Hence, it may alleviate the negative effect of loneliness and depressive symptoms, reducing the risk of suicidal ideation for nursing home residents. Therefore, future studies could consider the pattern of caregiver-older adults for resilience intervention in nursing homes. By enhancing the quality of psychological care ability from caregivers, the impact of risk factors on the suicidal ideation of nursing home residents can be hopefully alleviated.

## 5. Conclusions

This current study explored the possible development paths of loneliness to suicidal ideation and examined the protective mechanism of resilience in moderating the effect of loneliness and depressive symptoms on suicidal ideation among nursing home residents. The findings in the multiple moderating roles of resilience indicated that more specific interventions may be needed to be developed to target resilience in addition to loneliness and depressive symptoms. When resilience is strengthened, the negative impact (e.g., increased suicidal ideation) of loneliness and depressive symptoms on nursing home residents would be attenuated through the moderation paths identified in this study. Future research in this field should focus on positive psychological interventions (e.g., resilience) to enhance the psychological adaptability of nursing home residents to reduce suicidal ideation.

### Limitation

Some limitations of the present study associated with the data should be noted. First, the data were collected from only one city, showing a limitation in terms of their representativeness of and generalization to other cities all over China. Multicenter studies with large samples should be conducted for further research. We did not employ a specific scale to measure the socioeconomic contexts of nursing homes, even though a stratified randomized sampling was used to control the different levels of economy of each district. Future studies should pay attention to this point so as to control the socioeconomic factors. The measures used in this study were self-reported, which might be prone to recalling and reporting bias. For a few older adults who were disabled and illiterate, oral measurement was employed. Two different methods of questionnaire administration may somewhat influence the response rate, despite the fact that we adopted some quality control measures prior to and during the investigation (i.e., personnel training and assessment, unified guidelines). It is thus suggested future studies pay attention to this point so as to narrow the gap between the different measurements. Moreover, considering the cross-sectional nature of data, it is not feasible to examine the psychological transitions that could be influenced by major incidents. The association of variables about causation relationship is bound to involve some degree of speculation, as the data were measured simultaneously. Nevertheless, further longitudinal studies are still required to validate these relationships.

## Figures and Tables

**Figure 1 ijerph-18-05472-f001:**
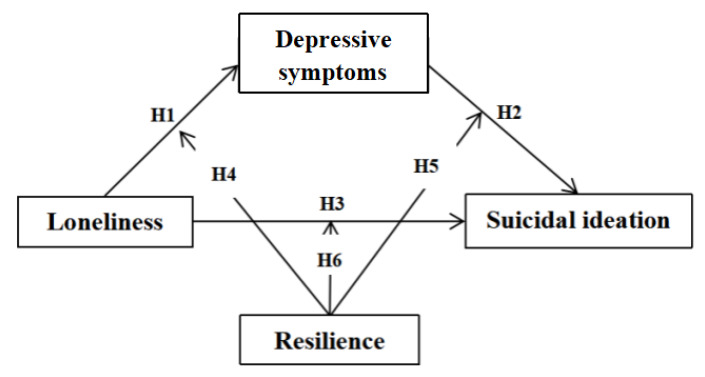
Hypothesis model.

**Figure 2 ijerph-18-05472-f002:**
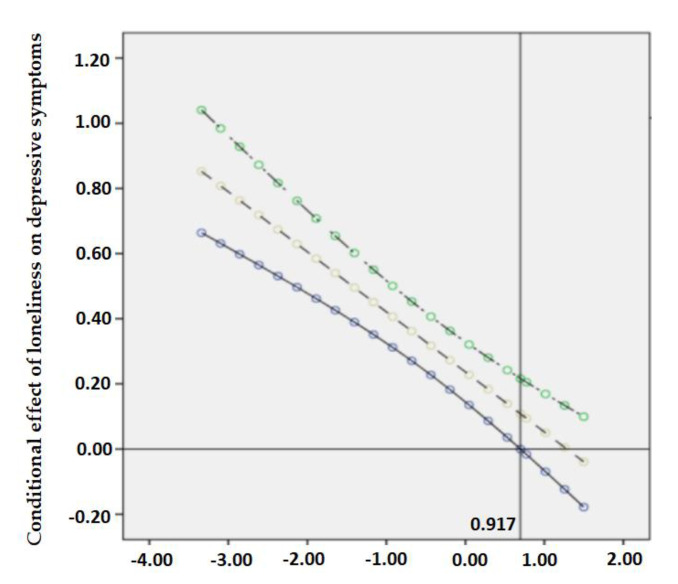
A plot of the effect of loneliness on depressive symptoms versus resilience, with confidence bands. * The curves above and below the line are the upper and lower 95%CI. The horizontal line denotes a conditional effect of zero. The vertical line represents the boundary of the region of significance, similarly hereafter.

**Figure 3 ijerph-18-05472-f003:**
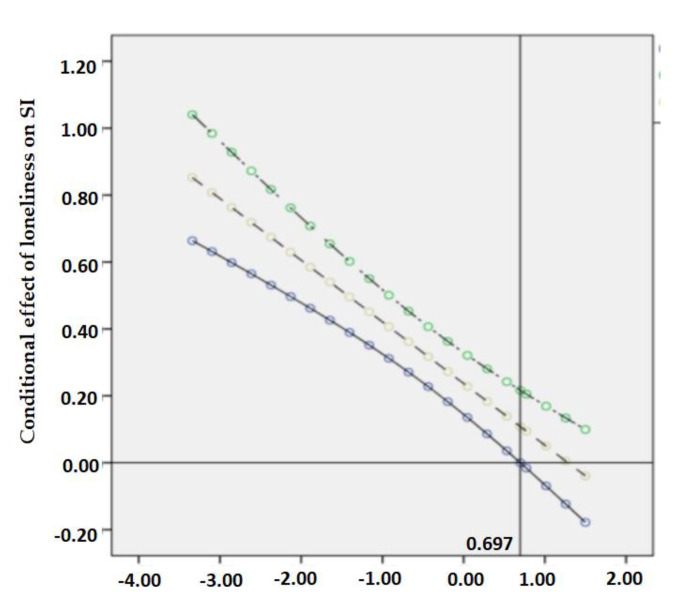
A plot of the effect of loneliness on SI at values of resilience.

**Figure 4 ijerph-18-05472-f004:**
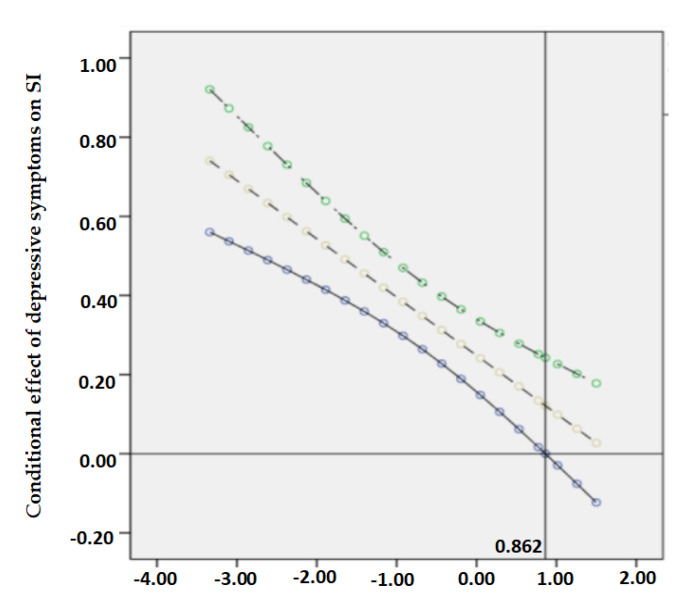
A plot of the effect of depressive symptoms on SI at values of resilience.

**Table 1 ijerph-18-05472-t001:** Basic characteristics and comparison between older adults with and without suicidal ideation (*n* = 538).

Variables	NSI (*n* = 458)	SI (*n* = 80)
Gender		
Female	260 (56.8)	61 (76.3)
Male	198 (43.2)	19 (23.8)
Age		
60–74	137 (29.9)	19 (23.8)
75–89	276 (60.3)	52 (65.0)
90–99	45 (9.8)	9 (11.2)
Marital status		
Unmarried	35 (7.6)	2 (2.5)
Married	103 (22.5)	9 (11.3)
Divorced	9 (2.0)	3 (3.7)
Widowed	311 (67.9)	66 (82.5)
Education		
Illiterate	191 (41.7)	49 (61.3)
Primary school	124 (27.1)	11 (13.7)
Junior high school	68 (14.8)	5 (6.3)
Senior high/above	75 (16.3)	15 (18.7)
Self-rated financial status		
Good	178 (38.9)	19 (23.8)
Medium	224 (48.9)	37 (46.2)
Poor	56 (12.2)	24 (30)
Children visit frequency		
Once per 1–2 weeks	282 (61.6)	40 (50.0)
Once per over 2 weeks	176 (38.4)	40 (50.0)
Number of physical diseases		
0	54 (11.8)	10 (12.5)
1–3	364 (79.5)	49 (61.3)
4–7	40 (8.7)	21 (26.2)
MMSE scores	22.74 ± 5.85	19.78 ± 6.00
Active suicidal ideation		65 (81.3)
Passive suicidal ideation		15 (18.7)

NSI = nursing home residents without suicidal ideation, SI = nursing home residents with suicidal ideation. Continuous variables are presented as mean and standard deviation (SD); categorical variables are presented frequency (*n*) and percentage (%).

**Table 2 ijerph-18-05472-t002:** Bivariate correlation between main variables.

	1	2	3	M ± SD
Loneliness	-	-	-	41.00 ± 11.44
Depressive symptoms	0.532 ***	-	-	5.21 ± 4.31
Resilience	−0.615 ***	−0.551 ***	-	27.61 ± 8.27
Suicidal ideation	0.484 ***	0.500 ***	−0.471 ***	2.54 ± 6.50

*** *p* < 0.001.

**Table 3 ijerph-18-05472-t003:** Mediation effect of depressive symptoms (*n* = 538).

	R^2^	F	B	SE	t	LLCI	ULCI
Outcome: Depression symptoms	0.343	46.230					
Loneliness			0.477	0.037	13.008 ***	0.386	0.564
Outcome: Suicidal ideation	0.326	36.579					
Loneliness			0.308	0.043	7.222 ***	0.200	0.416
Depression symptoms			0.325	0.044	7.403 ***	0.207	0.443

*** *p* < 0.001. Abbreviations: B: beta, regression coefficient; SE: standard error; CI: confidence interval; LLCI: lower limit confidence interval; ULCI: upper limit confidence interval.

**Table 4 ijerph-18-05472-t004:** Moderating effect of resilience on the mediation model (*n* = 538).

Outcome: Depression Symptoms					
**Variable**	**B**	**SE**	**t**	**LLCI**	**ULCI**
Loneliness	0.243	0.045	5.438 ***	0.155	0.331
Resilience	−0.328	0.045	−7.277 ***	−0.417	−0.240
Loneliness × Resilience	−0.133	0.027	−4.869 ***	−0.187	−0.079
**Outcome: Suicidal Ideation**					
**Variable**	**B**	**SE**	**t**	**LLCI**	**ULCI**
Loneliness	0.189	0.047	4.026 ***	0.097	0.282
Depressive symptoms	0.198	0.047	4.220 ***	0.106	0.290
Resilience	−0.124	0.049	−2.545 ***	−0.221	−0.028
Loneliness × Resilience	−0.109	0.035	−3.145 ***	−0.177	−0.041
Depressive symptoms × Resilience	−0.077	0.036	−2.170 ***	−0.147	−0.007

Controlling for gender, marital status, education, self-rated financial status, numbers of physical diseases, and MMSE; *** *p* < 0.001; Abbreviations: B: beta, regression coefficient; SE: standard error; CI: confidence interval; LLCI: lower limit confidence interval; ULCI: upper limit confidence interval.

## Data Availability

The data that support the findings in this study are available from the corresponding author, upon reasonable request.

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
