# Peer review of "How Loneliness Worked on Suicidal Ideation among Chinese Nursing Home Residents: Roles of Depressive Symptoms and Resilience"

_ijerph, 2021, doi:10.3390/ijerph18105472_

Round 1

Reviewer 1 Report

This authors of this manuscript aimed to test a moderated-mediation model to determine if the relationship between loneliness and suicidal ideation is mediated by depressive symptoms, and if resilience moderates each path in this mediation model. They found support for this model and suggested that resilience has a buffering effect on factors contributing to suicidal ideation. I appreciate the unique sample that the authors used in this study. There were, however, several major concerns that I believe must be addressed before this manuscript is ready for publication.

Major Issues:

  • The writing and grammar in this manuscript made it difficult for me to understand what the authors were trying to convey. This was particularly problematic in the introduction and the discussion sections and I had trouble understanding why this study is needed and how it is different from existing research on this topic. In the discussion, the authors presented research that seem to be related to their findings but I would appreciate it if the authors can contextualize their findings in the existing research and provide more explanation on how do these findings fit into the bigger picture.
  • The results section was also difficult to follow. There is a lot here and it may be helpful for the authors to streamline the written results and to use tables and figures more to help them declutter the text. (e.g. the basic characteristics of participants – many of that information can probably be included as a column “total sample” in Table 1).
  • The authors mentioned an interview process in their methods but the study was a self-report questionnaire. How did the interview fit into the procedures? Also, how did the researchers collect data when the participants were illiterate?
  • It would be important for the authors to explain how they determined their inclusion/exclusion criteria and why this criteria specifically.

Minor issues:

  • In the introduction, I believe that the authors did not accurately describe the 3 Step Theory (3ST) which is separate from the “ideation-to-action” framework which suggests that the factors contributing to suicidal ideation are different from those contributing to suicidal behavior.
  • The authors should present psychometric information for each measure in the methods section
  • It would be helpful to include additional detail on the Johnson-Neyman technique and support for using it in this study
  • The authors should avoid using causal language given that this is a cross-sectional study

Reviewer 2 Report

Thank you for inviting me to review this manuscript. The manuscript assesses the mediating role of depressive symptoms, and the moderating effect of resilience on the risk factors of suicidal ideation to attenuate the adverse contribution among nursing home residents. The study sample came from 538 home residents in a cross‐sectional study. The findings indicated the protective effect of resilience in alleviating the negative influence of risk factors for suicidal ideation, suggesting that positive psychological interventions for resilience building might be effective in suicide prevention among nursing home residents.

Keypoint 

  • The main concern is the innovation and significance of this study. It is unclear how the study results add value to the existing literature 
  • The theoretical framework of this study is not clear, although the author offered many frameworks.
  • Given that the study sample is not nationally representative. How the study results applied to the places with similar contexts or cultures? This is not discussed 
  • The English writing style shall be improved, there are several places with grammar errors. 

Abstract 

  • Relevant statistics associated with the results (e.g., coefficients of direct, indirect results, and point estimates) shall be reported 

Introduction 

  • The ‘ideation-to-action’ framework may not be relevant, as this study
  • The introduction introduced a lot of theoretical frameworks. It is unclear how they integratively guide the study hypotheses
  • Some of the hypotheses have been studied and documented in numerous studies. But not the innovation of this study 
  • Medical Disorders Scale was later introduced in the Methods, but it is unclear why the authors transition to measure the medical disorder. The hypothesis related to medical disorders is not discussed either. 
  • What makes the nursing homes in china different from other countries? Can the authors add more culturally relevant factors that make this study unique?

Method 

  • Design: The criteria for recruitment of the study sites are not clear. 
  • Design: what are the socioeconomic contexts of the nursing homes? How is the capacity of nursing homes? These are important factors related to suicidal ideation but not reported. 
  • Data collection: “answer a package of questionnaires during the interviews conducted in their single rooms” -- what is the package of questions?
  • Measures: Although the study uses standard scales. The author did not report or test the validity and reliability of the translated version. The authors did not report the psychometric properties of the Chinese measures among the elderly in this study sample 
  • Statistical analysis: the authors did not sufficiently discuss their approach of dealing with missing data and attrition
  • Statistical analysis: page 5 - is “Model 59” a typo?

Results  

  • Figure 2 has a max y-axis of “.00”, which is not possible?
  • “Conditional effect and confidence bands of resilience”: resorts of results are confusing. It is unclear what each of Figures 2-4 represent

Discussion   

  • The first paragraph, the author mention the study findings may have a theoretical foundation implication, while it is unclear which theory it is based on 
  • Since this is a cross-sectional study in China, the authors may want to discuss the results more related to Chinese culture.
  • One big concern is that the resilience scale is originally developed for Western adults. It is not clear how the scale performs when applied to the Chinese elderly.
  • Limitation: the authors did not address the sample selection, measures validity, and representation of the study.
  • The authors did not offer what kinds of clinical interventions can be derived from the results. 1-2 examples can help the readers understand.
  • There were no discussion on the mechanisms that may confound the results 

Tables 

  • The authors may need to consider adjusting the decimal points. Also, format Table 1 clearly with left alignment. 

Reviewer 3 Report

Thank you for  the authors to selecting such a important title. Later life experiences at nursing home are very emotional and critical situation for old age people.

Its recommended to authors to revise the following:

Introduction

  • Well written, however, Resilience part showed the relationship with depressive symptoms. It would be recommended to add some detail about resilience related to suicidal ideation.
  • This paper presents a thoughtful hypothesis model.

Methods:

  • Resilience : The 10‐item Connor‐Davidson Resilience Scale (CD‐RISC‐10): Is it data collected by the original version or Chinese translated version? Please write the deatails.

Results

  • Recommended to follow the standard terms: More information is presented in Table 1. [Table 1 near here]
  • Mediation effect of depressive symptoms: Please show the tables in below the result part (Example: Table 3 and 4) for easy understanding.
  • Recommended to write the legends of the tables (LLCI, ULCI B , SE etc)

 Discussion

  • Revise the spelling and terms (3line – depressive symptms – symptoms)
  • Need to revise the last couple of sentences of discussion: When resilience increased to a certain value, where 95%CI contained 0, the association among loneliness, depressive symptoms and suicidal ideation…..
  • poor physical health of older adults ……..

Limitations: To be after the conclusions - Revise the format

Conclusions

  • Recommended to add authors suggestion or recommendation about prevention of suicidal ideation and depreesive symptoms
